# Challenges of COVID-19 Case Forecasting in the US, 2020–2021

Velma K. Lopez[1]*, Estee Y. Cramer[2], Robert Pagano[3], John M. Drake[4], Eamon B. O'Dea[4], Madeline Adee[5], Turgay Ayer[6], Jagpreet Chhatwal[7], Ozden O. Dalgic[8], Mary A. Ladd[5], Benjamin P. Linas[9], Peter P. Mueller[7], Jade Xiao[6], Johannes Bracher[10], Alvaro J. Castro Rivadeneira[2], Aaron Gerding[2], Tilmann Gneiting[11], Yuxin Huang[2], Dasuni Jayawardena[2], Abdul H. Kanji[2], Khoa Le[2], Anja Mühlemann[12], Jarad Niemi[13], Evan L. Ray[2], Ariane Stark[2], Yijin Wang[2], Nutcha Wattanachit[2], Martha W. Zorn[2], Sen Pei[14], Jeffrey Shaman[14], Teresa K. Yamana[14], Samuel R. Tarasewicz[15], Daniel J. Wilson[15], Sid Baccam[16], Heidi Gurung[16], Steve Stage[17], Brad Suchoski[16], Lei Gao[18], Zhiling Gu[13], Myungjin Kim[19], Xinyi Li[20], Guannan Wang[21], Lily Wang[18], Yueying Wang[22], Shan Yu[23], Lauren Gardner[24], Sonia Jindal[24], Maximilian Marshall[24], Kristen Nixon[24], Juan Dent[25], Alison L. Hill[24], Joshua Kaminsky[25], Elizabeth C. Lee[25], Joseph C. Lemaitre[26], Justin Lessler[27], Claire P. Smith[25], Shaun Truelove[25], Matt Kinsey[28], Luke C. Mullany[28], Kaitlin Rainwater-Lovett[28], Lauren Shin[28], Katharine Tallaksen[28], Shelby Wilson[28], Dean Karlen[29], Lauren Castro[30], Geoffrey Fairchild[30], Isaac Michaud[30], Dave Osthus[30], Jiang Bian[31], Wei Cao[31], Zhifeng Gao[31], Juan Lavista Ferres[31], Chaozhuo Li[31], Tie-Yan Liu[31], Xing Xie[31], Shun Zhang[31], Shun Zheng[31], Matteo Chinazzi[32], Jessica T. Davis[32], Kunpeng Mu[32], Ana Pastore y Piontti[32], Alessandro Vespignani[32], Xinyue Xiong[32], Robert Walraven[33], Jinghui Chen[34], Quanquan Gu[34], Lingxiao Wang[34], Pan Xu[34], Weitong Zhang[34], Difan Zou[34], Graham Casey Gibson[30], Daniel Sheldon[2], Ajitesh Srivastava[35], Aniruddha Adiga[23], Benjamin Hurt[23], Gursharn Kaur[23], Bryan Lewis[23], Madhav Marathe[23], Akhil Sai Peddireddy[36], Przemyslaw Porebski[23], Srinivasan Venkatramanan[23], Lijing Wang[37], Pragati V. Prasad[1], Jo W. Walker[1], Alexander E. Webber[1], Rachel B. Slayton[1], Matthew Biggerstaff[1], Nicholas G. Reich[2], Michael A. Johansson[1]

1 COVID-19 Response, Centers for Disease Control and Prevention, Atlanta, Georgia, United States of America, 2 University of Massachusetts, Amherst, Amherst, Massachusetts, United States of America, 3 Unaffiliated, Tucson, Arizona, United States of America, 4 University of Georgia, Athens, Georgia, United States of America, 5 Massachusetts General Hospital, Boston, Massachusetts, United States of America, 6 Georgia Institute of Technology, Atlanta, Georgia, United States of America, 7 Massachusetts General Hospital, Harvard Medical School, Boston, Massachusetts, United States of America, 8 Value Analytics Labs, Boston, Massachusetts, United States of America, 9 Boston University School of Medicine, Boston, Massachusetts, United States of America, 10 Chair of Econometrics and Statistics, Karlsruhe Institute of Technology, Karlsruhe, Germany, 11 Heidelberg Institute for Theoretical Studies, Heidelberg, Germany, 12 Institute of Mathematical Statistics and Actuarial Science, University of Bern, Bern, Switzerland, 13 Iowa State University, Ames, Iowa, United States of America, 14 Mailman School of Public Health, Columbia University, New York, New York, United States of America, 15 Federal Reserve Bank of San Francisco, San Francisco, California, United States of America, 16 IEM, Bel Air, Maryland, United States of America, 17 IEM, Baton Rouge, Louisiana, United States of America, 18 George Mason University, Fairfax, Virginia, United States of America, 19 Kyungpook National University, Bukgu, Daegu, Republic of Korea, 20 Clemson University, Clemson, South Carolina, United States of America, 21 College of William & Mary, Williamsburg, Virginia, United States of America, 22 Amazon, Seattle, Washington, United States of America, 23 University of Virginia, Charlottesville, Virginia, United States of America, 24 Johns Hopkins University, Baltimore, Maryland, United States of America, 25 Johns Hopkins Bloomberg School of Public Health, Baltimore, Maryland, United States of America, 26 École Polytechnique Fédérale de Lausanne, Lausanne, Switzerland, 27 Gillings School of Global Public Health, University of North Carolina at Chapel Hill, Chapel Hill, North Carolina, United States of America, 28 Johns Hopkins University Applied Physics Lab, Baltimore, Maryland, United States of America, 29 University of Victoria and TRIUMF, Victoria, British Columbia, Canada, 30 Los Alamos National Laboratory, Los Alamos, New Mexico, United States of America, 31 Microsoft, Redmond, Washington, United States of America, 32 Laboratory for the Modeling of Biological and Socio-technical Systems, Northeastern University, Boston, Massachusetts, United States of America, 33 Unaffiliated, Davis, California, United States of America, 34 University of California, Los Angeles, Los Angeles, California, United

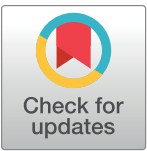

**Data Availability Statement:** The forecasts from models used in this paper are available from the COVID-19 Forecast Hub GitHub repository (https://github.com/reichlab/covid19-forecast-hub) and the Zoltar forecast archive (https://zoltardata.com/

project/44). The code used to generate all figures and tables in the manuscript is available in a public repository (https://github.com/cdcepi/Evaluation-of-case-forecasts-submitted-to-COVID19-Forecast-Hub).

**Funding:** National Science Foundation (NSF) grant DEB-2027786 supported EBO and JMD; NSF grant DMS-2027369, National Institute of Allergy and Infectious Diseases (NIAID) grant AI163023, and a gift from the Morris-Singer Foundation supported SP, JS, and TKY; NSF grants DMS-1916204, DMS-2203207, and CCF-1934884 and the Laurence H. Baker Center for Bioinformatics and Biological Statistics at Iowa State University supported XL, GW, LG, ZG, MK, LW, YW, and SY; NSF RAPID grants 2108526 and 2028604 supported MM, LG, SJ, and KN. Amazon Web Services/COVID-19 High Performance Computing Consortium supported JCL, ECL, JD, ALH, JK, JL, CPL, and ST. Swiss National Science Foundation, grant 200021172578, Fondo Integrativo Speciale Ricerca, grant EPIDOC, FISR-2020IP-04249, supported JCL; State of California supported JCL, JD, JK, ECL, JL, and ST; US Department of Health and Human Services (HHS) supported JCL, ALH, JK, ECL, JL, and ST; US Department of Homeland Security (DHS) supported JCL, JK, ECL, JL, and ST; Johns Hopkins Health System supported JD, JK, ECL, JL, and ST; Johns Hopkins University Modeling and Policy Hub supported JD, JK, ECL, JL, CPS, and ST; NSF grant 2127976 supported JD, JK, ECL, CPS, and ST; Los Angeles County Department of Public Health supported JD and ECL; US Centers for Disease Control and Prevention (CDC) grant 200-2016-91781 supported ALH, JK, JL, CPS, and ST; Office of the Dean at Johns Hopkins Bloomberg School of Public Health supported JK, ECL, JL, and ST; Los Alamos National Laboratory's Laboratory Directed Research and Development program grant 20200700ER supported LC, GF, IM, and DV; HHS/CDC grant 6U01IP001137 and HHS/CDC grant 5U01IP0001137 supported AV, MC, JTD, KM, APP, and XX; NSF grants 2027007 and 2135784 supported AS; NSF grant 1749854 supported GCB and DS; National Institutes of Health (NIH) Grant 1R01GM109718, NSF BIG DATA grant IIS-1633028, NSF grant OAC-1916805, NSF Expeditions in Computing grants CCF-1918656 and CCF-1917819, NSF RAPID grant CNS-2028004 and OAC-2027541, CDC grant 75D30119C05935, a grant from Google, University of Virginia Strategic Investment Fund award SIF160, Defense Threat Reduction Agency (DTRA) under Contract No. HDTRA1-19-D-0007, Virginia Dept of Health (VDH) contract VDH21-501-0141, Council of State and Territorial Epidemiologists/CDC grant 5 NU38OT000297,VDH contract VDH-

States of America, **35** University of Southern California, Los Angeles, California, United States of America, **36** Discreet Labs, Raleigh, North Carolina, United States of America, **37** New Jersey Institute of Technology, Newark, New Jersey, United States of America

* oko8@cdc.gov (VKL)

# Abstract

During the COVID-19 pandemic, forecasting COVID-19 trends to support planning and response was a priority for scientists and decision makers alike. In the United States, COVID-19 forecasting was coordinated by a large group of universities, companies, and government entities led by the Centers for Disease Control and Prevention and the US COVID-19 Forecast Hub (https://covid19forecasthub.org). We evaluated approximately 9.7 million forecasts of weekly state-level COVID-19 cases for predictions 1–4 weeks into the future submitted by 24 teams from August 2020 to December 2021. We assessed coverage of central prediction intervals and weighted interval scores (WIS), adjusting for missing forecasts relative to a baseline forecast, and used a Gaussian generalized estimating equation (GEE) model to evaluate differences in skill across epidemic phases that were defined by the effective reproduction number. Overall, we found high variation in skill across individual models, with ensemble-based forecasts outperforming other approaches. Forecast skill relative to the baseline was generally higher for larger jurisdictions (e.g., states compared to counties). Over time, forecasts generally performed worst in periods of rapid changes in reported cases (either in increasing or decreasing epidemic phases) with 95% prediction interval coverage dropping below 50% during the growth phases of the winter 2020, Delta, and Omicron waves. Ideally, case forecasts could serve as a leading indicator of changes in transmission dynamics. However, while most COVID-19 case forecasts outperformed a naïve baseline model, even the most accurate case forecasts were unreliable in key phases. Further research could improve forecasts of leading indicators, like COVID-19 cases, by leveraging additional real-time data, addressing performance across phases, improving the characterization of forecast confidence, and ensuring that forecasts were coherent across spatial scales. In the meantime, it is critical for forecast users to appreciate current limitations and use a broad set of indicators to inform pandemic-related decision making.

## Author summary

As SARS-CoV-2 began to spread throughout the world in early 2020, modelers played a critical role in predicting how the epidemic could take shape. Short-term forecasts of epidemic outcomes (for example, infections, cases, hospitalizations, or deaths) provided useful information to support pandemic planning, resource allocation, and intervention. Yet, infectious disease forecasting is still a nascent science, and the reliability of different types of forecasts is unclear. We retrospectively evaluated COVID-19 case forecasts, which were often unreliable. For example, forecasts did not anticipate the speed of increase in cases in early winter 2020. This analysis provides insights on specific problems that could be addressed in future research to improve forecasts and their use. Identifying the strengths

21-501-0135, NSF RAPID 2142997, VDH contract UVABIO610-GY23, VDH contract UVABIO619-GY24US, DOD contract SD00189-15-TO-1 supported AA, BH, GK, BL, MM, ASP, PP, SV, and LW; NIGMS grant (R35GM119582) and CDC (U01IP001122) supported EYC, AJCR, AG, ELR, AS, YW, NW, MWZ, and NGR; LANL-LDRD ER grant 20200700ER supported LC, GF, IM, and DO; NSF grants 2135784 and 2223933 supported AS. The content is solely the responsibility of the authors and does not necessarily represent the official views of any of the funding agencies. The funders had no role in study design, data collection, analysis, decision to publish, or preparation of the manuscript.

**Competing interests:** I have read the journal's policy and the authors of this manuscript have the following competing interests: APP report grants from Metabiota Inc outside the submitted work. J. S. and Columbia University declare partial ownership of SK Analytics. No other authors have competing interests to declare.

and weaknesses of forecasts is critical to improving forecasting for current and future public health responses.

## Introduction

Predicting the trajectory of an epidemic to support control and mitigation planning is the primary objective of infectious disease forecasting. To this end, large-scale, collaborative forecasting efforts across multiple disease systems, such as influenza [1–3], dengue [4], West Nile [5], and Ebola viruses [6], have been integrated into routine public health workflows and emergency response [7]. Researchers in academia, private institutions, and the United States (US) government built upon these frameworks to incorporate forecasts into the COVID-19 information systems used to inform pandemic response and created the US COVID-19 Forecast Hub. In April 2020, the US Centers for Disease Control and Prevention (CDC) and the COVID-19 Forecast Hub began collecting COVID-19 death forecasts [8]. Compared to death reports, case reports are a leading indicator of SARS-CoV-2 infections, as the time from infection to case report is typically shorter than that between infection and death report. Hence, information gleaned from case forecasts is potentially more actionable.

Case forecasts for all US counties (n = 3,143), states (n = 50), territories (n = 5), the District of Columbia (DC), and the nation as a whole were generated and collected beginning in July 2020, with ensemble forecasts of cases first posted on a CDC webpage on August 6, 2020 [8,9]. Due to public interest and their potential utility, case forecasts were also integrated into US government web pages and situational awareness updates [10]. In addition, COVID-19 case forecasts have been cited as useful for guiding personal risk-based decisions [11]. Because these forecasts potentially influence policies and personal decisions, accuracy and precision are of the utmost importance.

As part of routine use of the case forecasts in the COVID-19 response, real-time evaluation was conducted. One of the performance metrics included in the evaluation was the 95% prediction interval (PI) coverage, an estimate of the frequency at which the interval captures the eventually observed data. The 95% PI of a reliable forecast should capture eventually reported cases 95% of the time. However, the real-time evaluation indicated that case forecasts were not always reliable, with much lower 95% PI coverage than expected [12]. For example, in November 2020 as the 2020–2021 winter wave began, the 95% PI coverage for all states and territories was less than 50% for even the shortest, 1-week ahead forecasts from the ensemble–generally the most reliable forecast. Repeated periods of low coverage during subsequent surges led the CDC to stop posting COVID-19 case forecasts in December 2021. Though these forecasts showed poor performance, there are opportunities to develop more precise and reliable future predictions.

Evaluation of forecast performance provides an opportunity not only to assess prediction skill for the purposes of improving forecasts, but also to assess the reliability of the forecasts and foster transparency between forecast users and creators. While evaluation is recommended in forecasting research guidelines (i.e., EPIFORGE 2020 [13], a systematic review of COVID-19 models showed that half of published models did not include probabilistic predictions and that approximately one-fourth of published models did not include performance evaluations [14]. We have previously evaluated forecast performance of cumulative [15] and incident [16] COVID-19 deaths submitted to the COVID-19 Forecast Hub. Given that an ensemble of submitted models provided consistently accurate probabilistic forecasts at different scales in both evaluations, here we apply similar methods to assess the prediction skill of

the COVID-19 case forecasters, with particular interest in the COVIDhub ensemble model (that is, a model that combines predictions from forecasts submitted to the Forecast Hub). Specifically, we analyze prediction interval coverage and other aspects of nearly 10 million individual forecasts collected by the COVID-19 Forecast Hub for US jurisdictions between July 2020 and December 2021, the full period over which COVID-19 case forecasts were published by the CDC. We analyze relative forecast performance across spatial scales and phases of the pandemic to identify limitations and opportunities for future improvement of case forecasts.

## Results

### Summary of included team forecasts

A total of 14,960,171 forecasts were submitted by 67 teams throughout the analysis period (see S1 Appendix for submission patterns over time). Because forecasts were submitted at multiple geographic scales, we stratified analyses for 1) national forecasts, 2) state (including all 50 states), territory (US Virgin Islands and Puerto Rico), and DC forecasts), 3) county-level forecasts (including all 3,143 counties and county equivalents), split into five equal sized groups based on county population size.

We first assessed the locations, horizons, and time periods forecasted by each team to ensure that forecasts included all required quantiles and horizons and to limit comparisons to teams with substantial overlapping spatiotemporal coverage. Briefly, teams were included if they submitted the full range of required quantiles, included at least 50 of states/territories/DC or 75% of counties, and produced forecasts at least four weeks into the future for at least 50% of the time points in the study period. At the national level, 22 sets of team forecasts met these criteria (5,136 forecasts across dates and forecast horizons), 23 sets of team forecasts met the state/territory level criteria (280,132 forecasts across jurisdictions, dates, and forecast horizons), and 15 sets of team forecasts met the county-level criteria (9,415,460 forecasts across counties, dates, and forecast horizons). Overall, 64.8% of all submitted forecasts were included in the analysis (9,700,728 forecasts). Of the included forecasts, 11 sets of team forecasts met the inclusion criteria for analyzing submissions across all geospatial scales (8,125,220 forecasts for specific locations, dates and forecast horizons).

Each team included in the analysis submitted forecasts that were generated from unique model structures, data inputs, and assumptions (S1 Appendix). Two naïve models (the COVIDhub-baseline and CEID-Walk) and four ensemble models (the COVIDhub-4_week_ensemble, the COVIDhub-trained_ensemble, LNQ-ens1, and UVA-Ensemble), which combined multiple forecasts into one, were included in the 26 models evaluated (see Table A in S1 Appendix). The COVIDhub-baseline model projects the number of reported cases in the most recent week as the median predicted value for the next 4 weeks. CEID-Walk is a random walk model with a simple method for removing outliers. A total of seven models included data on COVID-19 hospitalizations, 12 models incorporated demographic data, and seven models used mobility data. Of the 26 evaluated models, three (COVIDAnalytics-DELPHI, CU-select, and UCLA-SuEIR) assumed that social distancing and other behavioral patterns changed during the prediction period.

The evaluation period consisted of 1–4 week ahead forecasts submitted in the 73 weeks from July 28, 2020, through December 21, 2021. Multiple phases of the US epidemic were included: the late summer 2020 increase in several locations, a large late-fall/early-winter surge in 2020/2021, the rise and fall of the Delta variant in the summer and fall of 2021, and the early phase of the Omicron variant's dominance in winter 2021 (Fig 1A). Performance of the national ensemble forecasts varied over this period (Fig 1B). For some forecasts, the median predictions were close to the cases eventually reported, and most reported numbers fell within

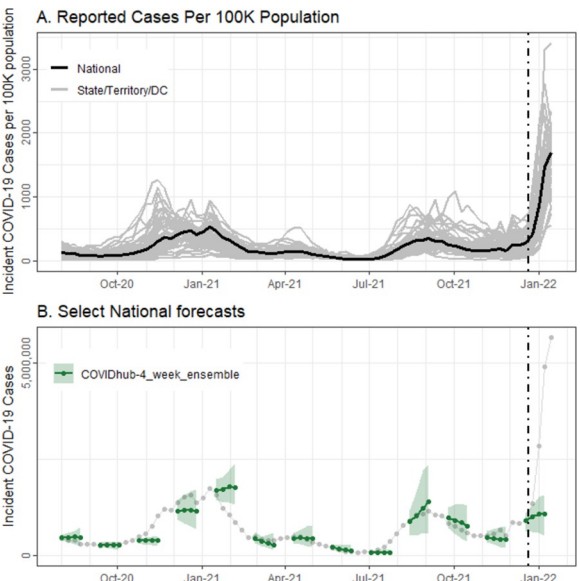

**Fig 1.** Weekly incident reported COVID-19 cases per 100K population, nationally (in black) and per state/territory/DC (in gray), over time in panel A. Panel B shows a subset of COVIDhub-4_week_ensemble forecasts (in green) over time, with the median predictions represented as lines and points and the 95% prediction intervals in bands. Reported incident cases (counts per week) are shown in gray. In both plots, the black, dashed vertical line shows the date that public communication of the case forecasts was paused.

the 95% PIs. However, forecasts made at other times, such as January 2021 or December 2021, diverged widely from the reported data. At those times, the forecasts missed substantial decreases and increases, respectively, with reported cases falling within the 95% prediction interval for only 1-week ahead forecasts.

## Aggregate performance

We evaluated aggregate forecast performance with two metrics: Weighted Interval Score (WIS), a proper score considering both precision and accuracy, and prediction interval coverage, an indicator of forecast uncertainty. Lower WIS values reflect forecasts with probability mass closer to observed values. We assessed scaled pairwise WIS relative to the baseline model (referred to throughout as relative WIS, or rWIS) for national and state/territory/DC forecasts (Fig 2). A rWIS less than one indicates performance that is better than the baseline model.

Overall, seven of 22 team's forecast models outperformed the COVIDhub-baseline model at the state/territory/DC level (i.e., had rWIS values less than 1.0), and 11 outperformed the baseline model at the national level. Six of these teams outperformed the baseline model at both scales: LNQ-ens1, COVIDhub-4_week_ensemble, USC-SI_kJalpha, LANL-GrowthRate, Microsoft-DeepSTIA, and CU-select.

Prediction interval coverage at the 95% level should be close to 95% for well calibrated forecasts. However, it was lower for most sets of team forecasts, with only one (LNQ-ens1) having coverage of at least 90% at all scales, while others were as low as 23%. PI coverage at 50% and 80% levels were also well below nominal levels for most sets of team forecasts, including the COVIDhub-4_week_ensemble (Fig 3). For the 50% prediction interval, no sets of team forecasts had coverage better than 36% at any scale. Only two sets of team forecasts had better coverage than the COVIDhub-4_week_ensemble for the geographic scales in which they submitted forecasts: LNQ-ens1 (all scales) and JHU_UNC_GAS-StatMechPool (state/territory/DC and large

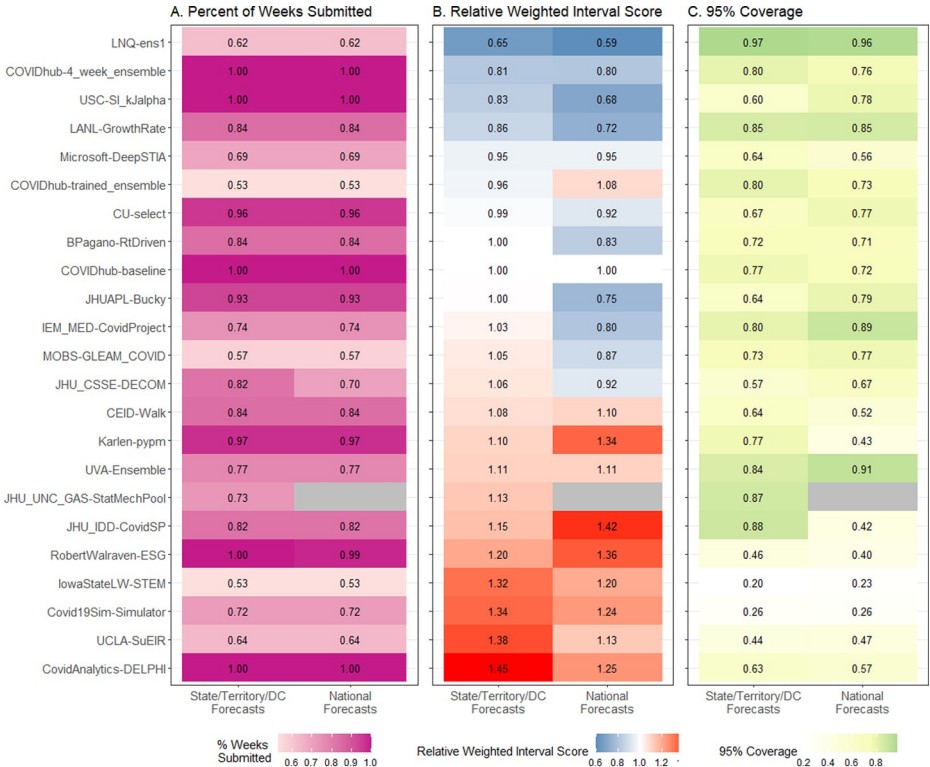

**Fig 2. Percent of weeks with complete submissions for all sets of team forecasts, scaled, pairwise relative Weighted Interval Score (rWIS; see _Methods_ for description), observed 95% prediction interval coverage, by geographical scale of submitted forecasts.** Teams are sorted by increasing state/territory/DC rWIS values.

county levels). We also found that calibration, WIS, and prediction interval coverage were all worse at 4-week horizons compared to 1-week horizons (S6 Appendix).

Forecast skill also showed distinct patterns across jurisdictional scales, with rWIS decreasing for larger jurisdiction scales (e.g., national vs. state/territory) or population sizes (e.g., larger counties vs. smaller counties, Fig 4) for most sets of team forecasts. In contrast to this general trend, for three sets of team forecasts, that pattern was inverted, one team had no distinct pattern, and the COVIDhub-4_week_ensemble had markedly consistent rWIS across all scales. Consistent with the aggregate findings, both LNQ-ens1 and COVIDhub-4_week_ensemble had rWIS lower than 1.0 at all scales, while LANL-GrowthRate had rWIS greater than 1.0 for smaller counties.

## Performance across jurisdictions

There was additional variability in forecast skill between jurisdictions. Only two team forecasts (LNQ-ens1 and COVIDhub-4_week_ensemble) performed as well as or better than the baseline for all included states and territories (Fig 5). Variation was higher between team forecasts than between specific jurisdictions, but the baseline model tended to outperform more models in some jurisdictions (e.g., the baseline was better in Colorado, Kansas, Puerto Rico) than in others (e.g., the baseline was worse in Mississippi, South Carolina, West Virginia). Spatial correlation is intrinsic to COVID-19 spread making correlation in the forecasts likely regardless of forecast skill. We found some evidence of spatial correlation in rWIS for many team forecasts (ensemble Moran's I: 0.36, p-value = 0.001, Fig A in S4 Appendix).

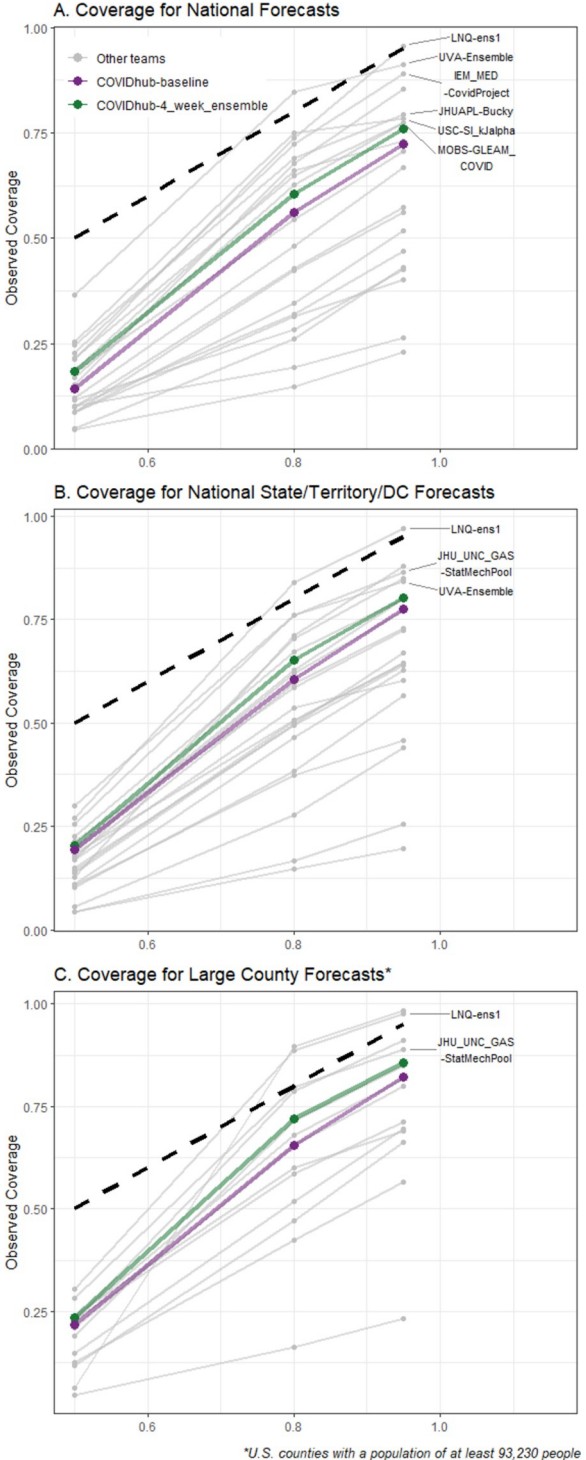

**Fig 3.** Expected and observed coverage rates for central 50%, 80% and 95% prediction intervals aggregated over time and horizon for national forecasts (panel A), state/territory/DC forecasts (panel B), and the largest county forecasts (panel C). The dashed line represents optimal expected coverage. Team forecasts that had closer to nominal coverage than the COVIDhub-4_week_ensemble model at all three coverage levels are labeled on the right side of the plots.

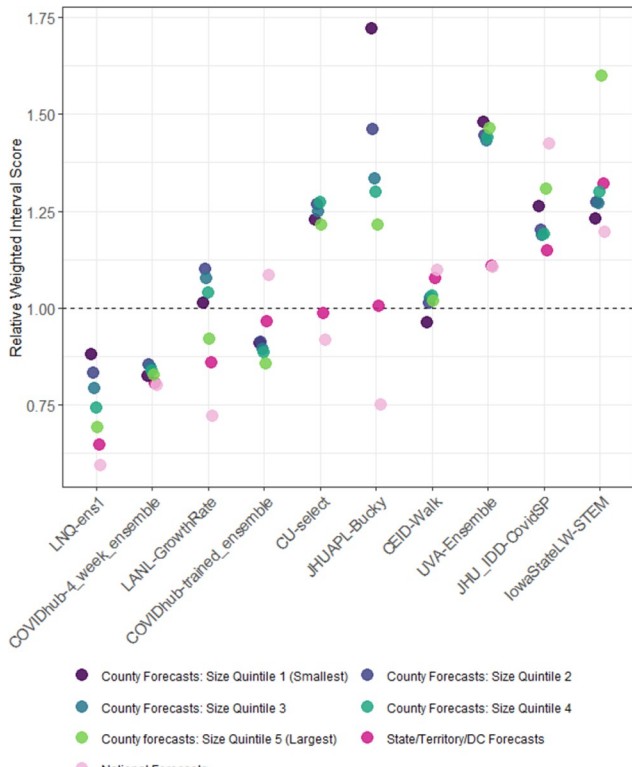

**Fig 4. Scaled, pairwise relative Weighted Interval Score (rWIS) (see _Methods_ for description) by spatial scale for sets of team forecasts that submitted forecasts for the US nation, states/territories/DC, and all US counties.** WIS is averaged across all horizons. The COVIDhub-baseline model has, by definition, a rWIS of 1 (horizontal dashed line). Teams are ordered by increasing state/territory/DC rWIS with the most accurate model on the left. Points for each team are staggered horizontally to show overlapping WIS values.

## Performance over time

WIS also varied over time (Fig 6). For example, all models had relatively high WIS in December 2020-January 2021 and low WIS in June 2021. Prediction interval coverage also varied between teams and over time, with most team forecasts exhibiting times of low coverage. Across most time points, the baseline model outperformed many team forecasts, including the COVIDhub-4_week_ensemble, though the ensemble more often outperformed the baseline in both WIS and prediction interval coverage at the national, state/territory, and large county scales. Increased WIS and decreased prediction interval coverage generally occurred with increasing case counts, such as in the fall of 2020 and summer of 2021. The worst performance was in the early Omicron wave in the winter of 2021. For the last set of ensemble forecasts posted by the CDC in December 2021 (https://www.cdc.gov/coronavirus/2019-ncov/science/forecasting/forecasts-cases.html), the WIS reached the highest level ever for all scales and the reported case numbers were outside the 95% prediction interval for most locations at every forecast horizon.

To further investigate these temporal patterns in performance, we first classified each forecasted week as _increasing_, _peak_, _decreasing_, or _nadir_ based on the estimated time-varying reproduction number for that given week and jurisdiction (Fig A in S5 Appendix). We then fitted Gaussian generalized estimating equations (GEE) models for each set of team forecasts, using a normalized, log transformed WIS value per forecast time and location as the model outcome. The regression models were adjusted for each prediction horizon and included a natural spline

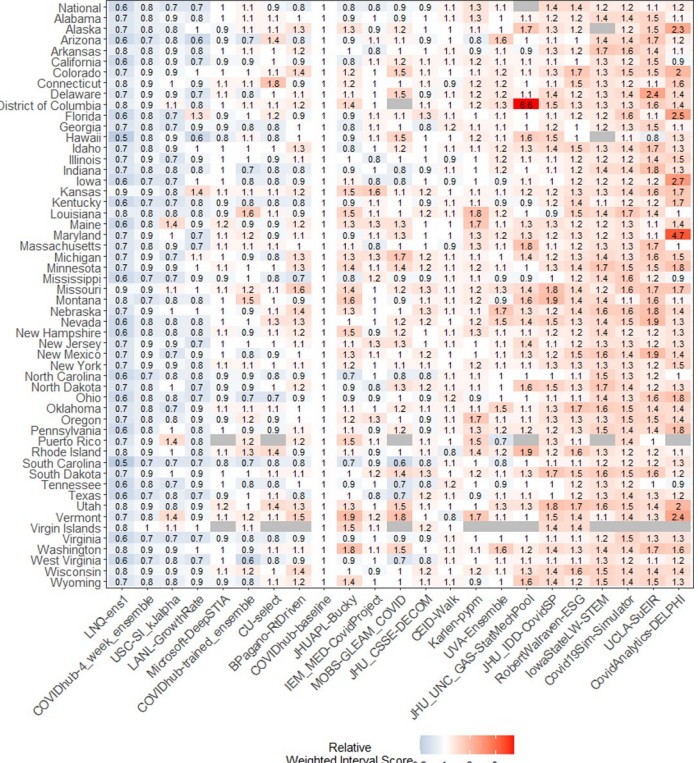

**Fig 5. Scaled, pairwise relative Weighted Interval Score (rWIS; see *Methods* for description) by location for national and state/territory/DC forecasts, averaged across all horizons through the entire analysis period.** National estimates are displayed first, followed by jurisdictions in alphabetical order. Team forecasts are ordered by increasing average state/territory/DC rWIS.

with two degrees of freedom for the time/state reported case counts to adjust for intrinsic increases in WIS due to higher values in reported cases (see S7 Appendix). In agreement with the aggregated results (Fig 2), we found that the expected WIS at the mean number of case counts across all jurisdictions was lower than the baseline for the better performing models (6 team forecasts and the ensemble) and higher than the baseline for others (8 team forecasts).

Forecast skill and coverage also varied across epidemic phases (Fig 7B and 7C). Compared to the baseline model across all phases, overall WIS for most models was better in nadir and peak phases and worse in increasing and decreasing phases. Likewise, 95% prediction interval coverage was highest in the nadir phase for nearly all teams while coverage in other phases was mostly lower than 95 percent. LNQ-ens1 and the COVIDhub ensemble had better WIS values than the baseline model in all epidemic phases between August 1, 2020, and January 15, 2022, with LNQ-ens1 also exhibiting close to nominal coverage across all phases.

We classified each forecast as increasing, decreasing, or stable or uncertain based on the 50% prediction interval relative to the most recent observed value. If the forecasts were able to correctly predict the direction of the epidemic phase, we would expect a high percentage of forecasts with an increasing trajectory to occur in an increasing epidemic phase, and likewise, a high percent of stable/uncertain trajectory forecasts in the peak and nadir phases and a high percent of forecasts with a decreasing trajectory in the decreasing epidemic phase. Increasing forecasts were most commonly made for the peak phase (predicting continued increases rather than a peak) followed by the increasing phase (Fig 8A), though overall the most common forecasts were for stable or uncertainty trajectories (Fig 8B). While the percent of decreasing forecasts for

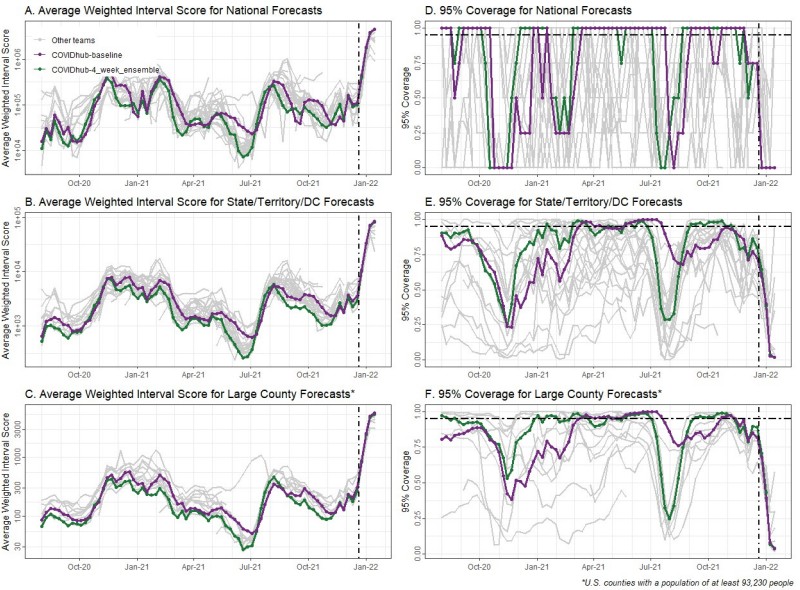

Panels A, D, B and E include: LNQ-ens1, Microsoft_DeepSTIA, COVIDhub-4_week_ensemble, USC-SI_kJalpha, CU-select, LANL-GrowthRate, JHU_CSSE-DECOM, COVIDhub-trained_ensemble, COVIDhub-baseline, Karlen-pypm, BPagano-RtDriven, JHUAPL-Bucky, UVA-Ensemble, IEM_MED-CovidProject, CEID-Walk, Covid19Sim-Simulator, IowaStateLW-STEM, UCLA-SuEIR, JHU_IDD-CovidSP, RobertWalraven-ESG, MOBS-GLEAM_COVID, and CovidAnalytics-DELPHI.

Panels C and F include: LNQ-ens1, COVIDhub-4_week_ensemble, CU-select, LANL-GrowthRate, COVIDhub-trained_ensemble, COVIDhub-baseline, JHUAPL-Bucky, UVA-Ensemble, CEID-Walk, JHU_UNC_GAS-StatMechPoole, IowaStateLW-STEM, JHU_IDD-CovidSP, UMass-MechBayes, FAIR-NRAR, FRBSF_Wilson-Econometric.

**Fig 6. Forecast accuracy over time, aggregated by geographic units, forecast horizon, and prediction date.** Panels A-C show average Weighted Interval Score (WIS); panels D-F show 95% prediction interval coverage. The black, dashed vertical line in all panels shows the date that public communication of the case forecasts was paused. The black, dashed horizontal line in panels D-F shows nominal 95% prediction interval coverage. National level forecasts are presented in A and D, state/territory/DC forecasts in B and E and large county forecasts in C and F.

the decreasing epidemic phase was high, forecasts often predicted declines for the nadir as well and some teams also frequently predicted decreases for the increasing phase (Fig 8C).

To examine whether our results were affected by reporting anomalies, we also conducted sensitivity analyses for data revisions, when data were revised at a later date, and for outlier data points, when reported cases were outside of weekly expected ranges (see S2 Appendix). We first identified weeks in which revised case counts as of April 2, 2022, differed from the case counts initially reported for that week, excluded them from the dataset, and reran the GEE models. With this partial dataset, the results were essentially unchanged. Next, we identified outliers as reported case counts outside of the expected range by at least two of the three following algorithms: a rolling median, a seasonal trend decomposition, and a seasonal trend decomposition without a seasonality term, each method over a 21-day window. Approximately 3% of weeks (686 of 27,489 total week-location combinations in the analysis period) had at least one day of reported cases identified as an outlier. We then excluded the weeks with outliers and the week following an outlier and reran the GEE models. This sensitivity analysis had comparable results to the models with the full data (see Fig C in S2 Appendix, Panel 1).

## Discussion

We evaluated performance of 9.7 million COVID-19 case forecasts at multiple geospatial scales in the US over approximately a year and a half. Real-time analyses and those presented here revealed important limitations in these forecasts. Forecast prediction intervals were largely over-confident, that is, prediction interval coverage was lower than the nominal value, particularly when case numbers were changing rapidly and forecasts could have been most useful. Few team forecasts outperformed a relatively simple and minimally informative baseline

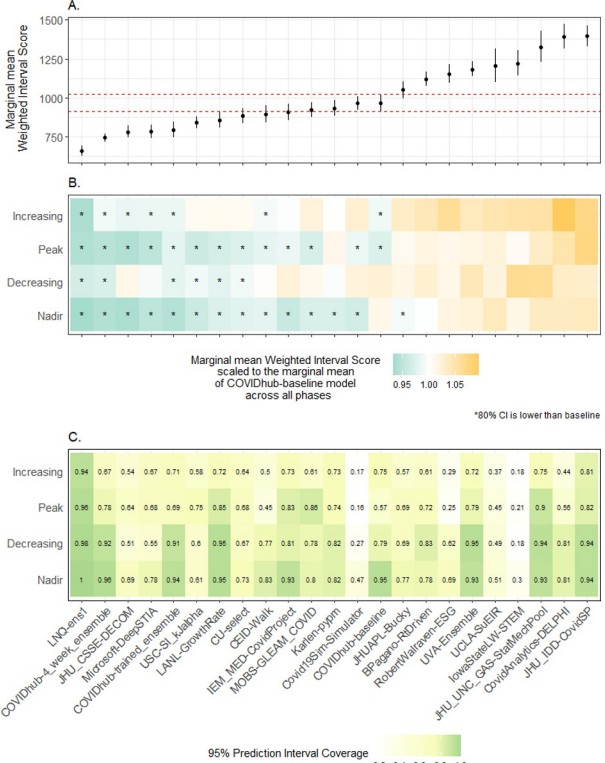

**Fig 7.** Estimated marginal mean Weighted Interval Score (WIS) and 95% confidence intervals for mean cases from team-specific GEE models for all 51 jurisdictions (Panel A). The 95% confidence intervals for the COVIDhub-baseline model are shown in dashed red vertical lines. Panel B presents each team's estimated marginal mean WIS per phase, scaled to the COVIDhub-baseline model's estimated marginal mean WIS for all epidemic phases. Teams with higher estimated marginal mean WIS values (i.e., greater than 1.0) are presented in shades of orange while teams with lower estimated marginal mean WIS (i.e., less than 1.0) are shown in shades of green. Forecasts for a team in a particular phase are marked with an asterisk (*) if the 80% confidence interval of the expected WIS outcome (normalized and on the log scale) was estimated by a model to be lower than the average expected WIS of the COVIDhub-baseline model across all phases. Panel C shows each team's mean 95% prediction interval coverage in each epidemic phase.

model. Forecast skill degraded for smaller geographic scales where forecasts could potentially be most useful. Forecast skill was also lowest when case counts were changing the most, in phases of increasing or decreasing transmission. These limitations of case forecasts indicate key areas for improvement and important reasons to use case forecasts with caution.

Several technical challenges for forecasts were evident in these analyses. First, cases are a relatively early indicator of transmission, with no clear leading signal in traditional public health surveillance data (e.g., unlike for death forecasts, where case counts themselves can provide information for predicting future deaths). While non-traditional data sources may provide a useful predecessor to changing population case counts, the evidence from previous work is unclear. For example, internet searches, medical claims, and online surveys have been used to modestly improve case forecast accuracy relative to models without those data [17]. Estimating case counts using both wastewater and clinical surveillance data has shown mixed results [18–21]. Additional integration of temporal dynamics could also be helpful. The case forecasts analyzed here were developed and evaluated based on the date when cases were reported, not when individuals were infected, became ill, sought care, or were tested. Additional detail on those dates could enable models to better capture the current dynamics using nowcasting approaches giving earlier signals of change.

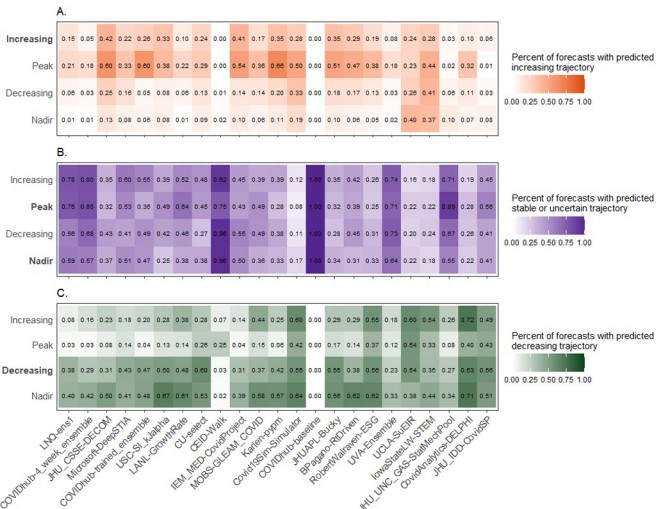

**Fig 8.** Percent of forecasts with predicted increasing trajectory per epidemic phase (Panel A), predicted stable or uncertain trajectory per epidemic phase (Panel B), and predicted decreasing trajectory per epidemic phase (Panel C). In each plot, epidemic phase labels are in bold when they correspond with the predicted direction of the forecast.

Second, and likely related to the challenge of cases being an early indicator, the models had substantial variation in skill between epidemic phases and between states. The baseline model performed relatively well in times of peaks and worse in nadirs because of relatively high uncertainty and in increasing and decreasing phases because it forecasted a flat trend. Comparing relative skill of forecasts in different phases with the mean baseline performance across all phases, we found that forecast skill was worst for the increasing phase followed by the decreasing phase for most teams. Classifying forecasted trajectories based on at least a 75% chance of increase or decrease, we found that a minority of forecasts for increasing and decreasing epidemic phases, had confident predictions of the trajectory and forecasts for peaks and nadirs were more likely to be increasing and decreasing forecasts, respectively. Teams generally underpredicted cases in the increasing phase and overpredicted in the decreasing phase. Underprediction also carried into the peak forecasts for most teams, although several teams overpredicted peaks. In many of the periods with high WIS (e.g., the 2020–2021 winter, Delta, and Omicron waves), the COVIDhub ensemble and other teams predicted possible or probable increases or decreases, but not at the rate that occurred. This effect may be even stronger than our results show as they rely on a comparison to the baseline which, by definition, does not predict change. While epidemic phase is unknown in real-time, it too can be estimated, and these results and others suggest that accounting for epidemic phase when making predictions could improve the forecast skill of ensemble models [22,23]. Additional data, as discussed above, or model components associated with distinct phases could also help improve predictive capabilities. Seasonal changes in transmission biology and human behavior, emergence of variants, and changing mitigation behavior all contribute to transmission dynamics. While some forecasting models incorporate seasonality and variants, only three models included some version of short-term behavior change and characterizing the interaction between behavior and transmission has lagged [24–26]. Nevertheless, even with the benefit of additional data, it is challenging to build transmission models that can capture all of the rapid change-points in cases, which were one of the foremost leading indicators. Ensemble approaches offer another opportunity to mitigate phase-specific differences. Team modeling skill across phases was highly heterogeneous, but two ensemble approaches were better than the baseline in all phases.

Another challenge across most forecasts was overconfidence, a pattern seen with other infectious disease forecasts [4,16]. The baseline model predicted a flat trend, yet it outperformed most team forecasts in the increasing and decreasing phase only because its predictions had higher uncertainty around that flat trend. The COVIDhub ensemble performance, on the other hand, benefitted by combining uncertainty across multiple models, yet, like the constituent models, also exhibited overconfidence. The temporal and phase-specific analyses suggest that it is, during rapid increases and decreases, that model overconfidence was most pronounced. Previous infectious disease forecasting work has shown that ensembles tend to have wider prediction intervals that are more likely to capture the eventually reported outcome and thus reduce overconfidence compared to their constituent models [4,16]. Wider prediction intervals, reflecting increased uncertainty, can mediate some impacts of overconfidence. However, forecasts would be most useful if they were both reliable and informative - that is, if they could accurately capture the uncertainty, while also providing more precise estimates, rather than merely increased uncertainty [27,28].

Finally, while forecasts would be most actionable at local scales, performance was generally worse for smaller than larger jurisdictions. Other infectious disease forecasting systems have found better forecast skill at smaller geographic scales, likely because local transmission dynamics (e.g., a county) are a better predictor of local than aggregate transmission (e.g., a state) [29]. We compared WIS across scales by comparison to the baseline model to adjust for missing forecasts and for WIS scaling relative to the magnitude of observed outcomes. After those adjustments, population size had a clear association with forecast skill, likely reflecting the relative role of stochastic dynamics. For better local forecasts, models may need to explicitly account for stochasticity. Forecasts could also be improved by better leveraging spatial information, such as dynamics in neighboring counties or nearest urban centers. Many of the forecasts here, including the top-performing ones, showed possible indications of spatial correlation in state-level performance, suggesting that spatial dynamics may not be accounted for fully. Local forecasts remain a key public health need, as local forecasts are more likely to reflect local conditions and motivate local mitigation action.

Overall, these findings, as well as the real-time evaluations, indicated that COVID-19 case forecasts were not reliable as a single indicator for pandemic response of a novel pathogen. Similar to other forecasting studies, we found that the ensemble was among the most reliable forecasts [3,4,16,30], outperformed only by LNQ-ens1 across the metrics evaluated here. Thus, while the overall best forecasts had poor performance at key times, other forecasts were often even worse at these same time points. Weighted (or trained) ensembles offer another potential avenue for improvement [31–33], but the version implemented here did not outperform the simple, median ensemble, likely reflecting limited historical data [34] and variation in team forecast submissions [35,36].

While COVID-19 deaths are a more lagging indicator of infections than case reports, and so may be less useful as an input to public health decision making, forecasts of deaths generally showed better forecast skill (e.g., most team death forecasts outperformed the baseline, ensemble rWIS was lower, and ensemble interval coverage was higher) [16]. Similarly, COVID-19 hospitalization forecasts in France have also shown high forecast skill [37]. Better performing US death and French hospitalization forecasts share one factor in common: models generally used local case reports as an input to inform their forecasts. While public health decision making should not rely on case forecasts alone, they may still be helpful in the context of other important indicators, such as the case, hospitalizations, and death data. Nowcasts and real-time estimates of the effective reproductive number can also provide insight into current dynamics [38–41]. Together, a suite of indicators is more informative for outbreak response than a leading indicator alone.

The analysis presented here includes important findings about real-time applied forecasting in an emerging pandemic to inform pandemic response rather than to address specific research aims of improving predictions. First and foremost, due to the submitted format of the forecasts and the overall goal of soliciting diverse models, our analysis was limited to WIS and prediction interval coverage. Other proper scoring metrics with different characteristics have been used in other challenges and may provide different insights [42–44]. Second, several factors limit the strength of our findings and ability to understand underlying mechanisms of predictive performance. Notably, we compared the forecasts to a changing record of reported cases. Throughout the COVID-19 outbreak, cases have been reported with jurisdiction- and time-varying delays and have been revised over time, resulting in varying forecast targets. In addition, the definition of a reported COVID-19 case also changed over time and varied between states. These changes were a result of many factors, including laboratory capacity and implementation of home-based testing, and may have affected forecast skill in other ways. Our sensitivity analyses found no qualitative differences in our main findings when we excluded forecasts for time points with revised data or when we excluded outlier data points. Nevertheless, forecasting teams were greatly impacted by the evolving landscape of COVID-19 case surveillance. More timely and consistent reports likely would improve both the process of making forecasts and forecast skill.

The overall goal of the COVID-19 Forecast Hub was to provide forecasts in near real-time for decision making. While the collaborative efforts of the Hub achieved this goal despite a changing epidemic landscape, nevertheless, the open nature of COVID-19 forecasting also limits understanding the drivers of forecast performance. Many teams participated at different times, some intermittently, and provided varied and limited descriptions of their forecast methods. While we were able to adjust our evaluation for differences in varying submissions, we are unable to assess the underlying impact of modeling approaches on performance since we do not have the granular details on forecast methods and how they evolved over time for all team forecasts. For example, the LNQ-ens1, which outperformed all other forecasts by most metrics, only submitted forecasts for approximately two-thirds of the analysis period and stopped in June 2021 (prior to the Delta wave). The model is described as a combination of three machine learning models, leveraging other embedded models and datasets, with weights that "are chosen by hand each week based on performance in the previous week" (see LNQ-ens1 metadata, https://github.com/reichlab/covid19-forecast-hub/blob/b12f916abc859bf59ea584b64f53afc2982042fd/data-processed/LNQ-ens1/metadata-LNQ-ens1.txt, at [45]). The ensemble approach used in the LNQ-ens1 model building likely contributed to the overall performance. However, several other ensemble models had lower performance than the LNQ-ens1 model; we are unable to assess whether LNQ-ens1 performance gains were due to a particular component model or dataset, the hand weighting procedure, or something else. The brief descriptions submitted to the COVID-19 Forecast Hub, such as for the LNQ-ens1, must include a summary of the methods used and may indicate a variety of unique features such as input data, parameters, model fitting, etc. [45]. However, the level of detail provided in these descriptions varies between teams, and we do not have enough information to determine which aspects of individual models were important determinants of forecast performance. To elucidate associations between modeling approaches and forecast skill, additional research is needed. Future work to support improved forecasting will require assessing the impact of specific features (e.g., through ablation analyses) using retrospective, stable data systems and retrospective evaluation of the full forecasting process (e.g., from data wrangling to final forecast production).

Infectious disease forecasting continues to present many challenges and opportunities for improving outbreak response. Forecasts should be leading indicators of future activity. While

the COVID-19 case ensemble forecasts were good leading indicators at many points in time, they were highly variable across teams and unreliable at longer horizons and during periods of rapid change. Case data were integrated in COVID-19 mortality forecasts, which proved to be more reliable, likely in part due to reported cases being leading indicators of reported deaths [15,46]. However, because deaths are a lagging indicator, death forecasts are less useful for short-term outbreak responses. Evaluation of the case forecasts provided insight on limitations of early forecasts and research avenues for improving them. These insights and the real-time forecasts provided by this effort were the product of large-scale collaboration between researchers and public health responders to confront the COVID-19 pandemic. Learning from and improving forecasting for COVID-19, other infectious diseases, and future pandemics will benefit from continuing and expanding these collaborative efforts.

## Methods

The US COVID-19 Forecast Hub [47] is a consortium of researchers that develop and share forecasts of COVID-19 reported cases, hospitalizations, and deaths with the goal of leveraging information from individual models that predict the near-term burden of COVID-19 in the United States. Teams that submitted models to the US COVID-19 Forecast Hub used a wide variety of methodologies and data (Table A in S1 Appendix). Beyond serving as a repository for forecasts, submitted data were also aggregated by scientists at the COVID-19 Forecast Hub to generate two models that we included in this analysis: the COVIDhub-4_week_ensemble and the COVIDhub-trained_ensemble. Since the beginning of the COVID-19 Forecast Hub, the quantile predictions from each week's submitted models were used as input data for the COVIDhub-4_week_ensemble. Ensemble aggregation methods evolved over time; for this analysis period, the ensemble forecast was calculated as the median across forecasts from all models at each quantile level. Additionally, beginning on February 1, 2021, the COVID-19 Forecast Hub also generated a weighted ensemble (COVIDhub-trained_ensemble). Models were selected for weighted ensemble inclusion based on their past performance over various window periods and given a weight prior to aggregation. The methodology evolved over time and details are available on the model's metadata file on the COVID-19 Forecast Hub GitHub repository (see *Data and code availability and reporting guidelines*).

The COVID-19 Forecast Hub, and death forecasts submitted to the Hub have been described in detail elsewhere [8,15,16]. The Hub's incident COVID-19 case forecasts, which were first solicited in July 2020, have similar submission requirements to the death forecasts. Important differences include an expanded geographical scale (national; state, territory, and DC; and county levels) and reduced number of required quantiles in the probability distribution (7 quantiles in total: 0.025, 0.10, 0.25, 0.50, 0.75, 0.90, and 0.975). Predictions for weekly incident COVID-19 cases can be submitted for up to 8 weeks in the future, although our analysis only includes predictions made for 1–4 weeks into the future.

We evaluated submitted forecasts between July 28, 2020, and December 21, 2021 (2020 epi week [EW] 31–2021 EW 51), which encompasses 74 weeks. Because forecasts were submitted at multiple geographic scales, we conducted separate analyses for 1) national forecasts, 2) state, territory, and DC forecasts, 3) county forecasts, and 4) sets of team forecasts for all three geographic scales. When appropriate, we compared forecast performance to that of a naïve model, created by the COVID-19 Forecast Hub, the COVIDhub-baseline. The COVIDhub-baseline model, created each week, was designed to be a neutral model to provide a simple reference point of comparison for all models. This baseline model forecasts a predictive median incidence equal to the number of reported cases in the most recent week, with uncertainty based on the empirical distribution of previous differences between the median and observed values [16].

### Inclusion criteria

Teams were included in the evaluation when they submitted forecasts with a complete set of quantiles for each 1- through 4-week ahead target predictions. Additionally, teams must have met the following inclusion criteria:

1. had predictions for at least 50 locations (states, territories, or DC) for the state, territory, and DC level analyses; and for at least 75% of counties included in each population size quintile per submission week for the county-level analyses;

2. had submissions for at least 50% of the weeks included in the analysis period per location forecasted.

Application of these inclusion criteria provided a more comparable set of forecasts for scoring with the attempt to reduce biased scores if teams only forecasted for a limited number of locations or number weeks. Teams meeting these inclusion criteria, and their submissions over time, are depicted in Figs A and B in S1 Appendix.

### Ground truth

Forecasts were evaluated against the reported COVID-19 case reports collated by the Johns Hopkins Center for Systems Science and Engineering (CSSE) [48]. To calculate weekly incident reported cases, we subtracted the cumulative count for each Saturday from the cumulative count for the next Saturday, such that each incident weekly count reflects the number of cases reported from Sunday through Saturday in a given week. We aggregated reported counts from smaller geographic units into their larger unit. For example, counts in a given state are the aggregate of the county-level reported counts and national counts are the sum of all states, territories, and DC.

CSSE reports data in real-time. Thus, data may be revised if the reporting health system makes public updates to its surveillance data. At times, such revisions may result in negative daily counts or in increases to case counts if the date of cases is shifted from one day to another or the definition of a reportable case is changed. We examined the percent change between the first reported cases in each state, DC, and territory per date relative to the counts in the surveillance file from April 2, 2022. We also assessed the influence of revised data on the final model outcomes (see S2 Appendix) and the presence of negative case counts in the timeseries. Less than 1 percent of time points in the analysis period had negative daily case counts in the largest US counties. Negative counts were observed at the state/territory level only twice: in Missouri during the week of April 17, 2021, and Virgin Islands during the week ending October 10, 2020. The state of Florida reported 0 cases on November 27, 2021. We excluded all weeks and locations with negative counts as well as the week with 0 incidence in Florida in our primary analyses.

Additionally, we also examined whether a reported case count was an outlier in the case trend for each state. Anomalies in case data trends have not been uncommon throughout the pandemic, as reporting entities have uploaded large batches of surveillance data on a single day. To assess whether cases were outside of the expected range of reported cases over time, we applied three outlier detection algorithms, each with a 21-day window: a rolling median, a seasonal trend decomposition, and a seasonal trend decomposition without a seasonality term. We then classified a given count as an outlier if it was detected as such by at least two of the three algorithms. Using these data, we ran several sensitivity analyses to assess the likely impact of anomalous data points on model performance. Sensitivity analyses examining the robustness of our findings to reporting anomalies are presented in S2 Appendix.

Additional information about the CSSE data, and revisions to the dataset, is publicly available on a GitHub repository:

https://github.com/CSSEGISandData/COVID-19/tree/master/csse_covid_19_data.

## Forecast locations

Forecasts for incident cases were submitted for the national level, 50 states, 5 territories (American Samoa, Guam, the Northern Mariana Islands, Puerto Rico, and the US Virgin Islands), the DC, and 3,142 US counties. We excluded two relatively new (2019) counties in Alaska (Federal Information Processing Standard code 02063 and 02066) because they were not included by most teams. Because fewer teams submitted forecasts for American Samoa, Guam, the Northern Mariana Islands, we excluded these territories from the analysis. Some teams treated DC as both a county and a jurisdiction, so we excluded DC from the county forecasts. In addition, because county population size and transmission are correlated and case counts and forecast performance are also correlated, we grouped counties into 5 quintiles based on their population sizes, with cut points at 8,908; 18,662; 36,742; and 93,230 people; most analyses used forecasts from the quintile with the largest population size (n = 628). We hypothesized that small counties would be more likely to have better forecast accuracy because they had zero or very few reported cases. We thus chose to stratify counties by size to minimize any bias from aggregation. Performance results for most county forecasts are presented in S3 Appendix, and state-level spatial correlation is presented in S4 Appendix.

## Defining epidemic phases

For every state and DC, we independently classified each forecast week based on the estimated time-varying reproduction number ($R_t$) for that given week. State-level $R_t$ estimates were obtained from https://github.com/epiforecasts/covid-rt-estimates [49]. We extracted the $R_t$ estimate for the Wednesday of each week from all available files. Because $R_t$ estimates were updated on a rolling basis in near real-time, there were multiple estimates generated for the same date; we calculated the median estimated $R_t$ per date for the upper and lower 90% credible interval and the median value (August 1, 2020 –January 15, 2022, or 2020 EW 31–2022 EW 2, reflecting 77 weeks in total). Each forecast week was then classified into one of the following categories based on the $R_t$ estimates: *increasing*, *peak*, *decreasing*, and *nadir*.

*Increasing* and *decreasing* phases reflect weeks in which $R_t$ had a 90% probability of being greater than or less than 1.0, respectively. There were several periods of rapid transmission in certain jurisdictions where $R_t$ dipped above/below the 1.0 threshold but did not remain on an upward or downward trajectory. Thus, we classified weeks between two increasing phases as *increasing* and weeks between two decreasing periods as *decreasing*. Weeks between increasing and decreasing phases were classified as *peaks*, whereas *nadirs* were defined as periods between decreasing and increasing phases. Periods at the beginning or the end of an analysis period were classified as a continuation of whichever phase preceded or followed them. The proportion of weeks classified as each epidemic phase and graphical depictions of $R_t$ are provided in S5 Appendix. The proportion of weeks suggests non-stable Rt trajectories in most locations and there is general concordance between $R_t$ and reported cases.

## Evaluation methodology

We evaluated probabilistic forecast accuracy using two different metrics, empirical prediction interval coverage rates and weighted interval scores (WIS) [43]. Coverage was calculated by determining the frequency with which the prediction interval contained the eventually observed outcome for the 50%, 80% and 95% intervals. WIS reflects a weighted estimate of

sharpness (i.e., the range of the predicted interval) and calibration (i.e., precision or error) across the three prediction intervals and the median prediction, with higher WIS and indicating lower forecast skill. WIS also integrates measures of overprediction and underprediction, that is, the difference in the observed value and the lower or upper limit of the prediction interval. Importantly, WIS is highly correlated with the magnitude of observed and forecasted values. We used mean absolute WIS to assess forecast accuracy over time and mean relative WIS (rWIS) to assess forecast accuracy over space. Relative WIS was estimated by calculating the geometric mean of WIS across all sets of team forecasts and scaling that value to the WIS of a naïve model, the COVID-hub baseline. This approach eases interpretation, where values greater than 1.0 reflected worse accuracy than the baseline model and values below 1.0 reflected better model performance. Additionally, the pairwise relative comparison helps account for missing forecasts. Both coverage and WIS have been described in detail elsewhere [16,43]. Horizon-specific results for national, state/territory/DC, and large counties are presented in S6 Appendix.

To assess the association between WIS and epidemic phase for each team, we fitted separate Gaussian generalized estimating equation (GEE) models per team (Eq 1) with an independent working correlation structure at the state level. This structure assumes that observations are correlated within a state (denoted as $l$ in the equations below), but not correlated over time in said state. Cases and weighted interval scores were log transformed and then standardized (subtracting the mean and dividing by the standard deviation) prior to fitting the model, as this transformation yielded more computationally and numerically stable estimates. We define those resulting variables as stdWIS and stdCases. The expected value for a standardized WIS for time ($t$) and location ($l$), with forecasts from a given team's model, is as follows:

$$log\left(stdWIS_{t,l,h}\right) = \beta_0 + \alpha_{p[t,l]} + \gamma_h + ns\left(log\left(stdCases_{t,l,h}\right)\right) + \varepsilon_{t,l,h} \qquad (1)$$

Where $p[t,l]$ is an index that reflects the phase of each time ($t$) and location ($l$), $h$ is the horizon of the forecast in weeks, and $ns(\cdot)$ represents a natural spline with two degrees of freedom. The model intercept is represented by $\beta_0$ and error by $\varepsilon_{t,l,h}$. Using a regression model allows us to summarize patterns of overall average performance between teams while accounting for high correlation and variation in the scores. Comparisons of rWIS, in contrast, do not allow for formal inference with statistical hypothesis testing or interval-based inference. Prior to applying this regression model structure, our model building approach included exploratory analysis of several structures appropriate for longitudinal analysis. We examined model residuals, influential observations, goodness of fit metrics, and the impact of changing the functional form of the variables included in the model.

The inclusion of reported cases in models permitted flexible adjustment for the wide range in cases between and within jurisdictions, which led to a wide range of possible WIS values, as WIS values tend to be higher when counts are higher. Expected WIS values were computed by first obtaining a marginal mean from the GEE model and then undoing the transformations by exponentiating and un-standardizing the marginal mean. This was done separately for each team for all phases and for each team and each phase individually (see S7 Appendix for estimated team-specific marginal mean WIS relative to reported case counts). Additionally, we calculated whether the 80% confidence interval (based on Gaussian distributional assumptions) for each team's expected WIS outcome (on the log-scale and normalized, as described above) was less than the average baseline model across all phases (i.e., the marginal mean WIS for the baseline model).

To determine the direction of the forecast predictions, each model's 50% PI was compared to the last known incidence value. Forecasts were categorized as increasing predictions when

the lower limit of their 50% PI was greater than the last known value, decreasing predictions when the upper limit of their 50% PI was less than the last known value, and stable/uncertain if their 50% PI contained the last known value.

All analyses were conducted using the R language for statistical computing (v 4.0.3) (50), and the following packages were used for the main analyses: *scoringutils* (44), *covidhubUtils* (51), *geepack* (52). Additionally, we included the EPIFORGE 2020 reporting guideline checklist in S8 Appendix to indicate each page in this evaluation that corresponds to each specific recommendation (13).

This activity was reviewed by the CDC and was conducted consistent with applicable federal law and CDC policy. See e.g., 45 C.F.R. part 46, 21 C.F.R. part 56; 42 U.S.C. §241(d); 5 U.S. C. §552a; 44 U.S.C. §3501 et seq.

**CDC disclaimer**: The findings and conclusions in this report are those of the authors and do not necessarily represent the official position of the Centers for Disease Control and Prevention.

## Supporting information

**S1 Appendix. Team submissions, methods, and data.** Fig A and B. Forecasts submitted over time at the national, state-territory-DC level in Fig A and at the county scale in Fig B. The number of forecasted locations submitted each week nationally or at the state, territory and DC level is included, while the county level forecast submissions show the percent of counties per quantile that were submitted each week. Sets of team forecasts meeting the inclusion criteria for this main analysis are labeled with an asterisk (*). Table A. List of models evaluated, including sources for case, hospitalization, death, demographic, and mobility data when used as inputs for the given model. We evaluated 26 models contributed by 24 teams. The COVID-hub team submitted three models including the baseline model and the ensemble model. A brief description is included for each model, with a reference where available. The last column indicates whether the model made assumptions about how and whether social distancing measures were assumed to change during the period for which forecasts were made.
(DOCX)

**S2 Appendix. Revision and outlier sensitivity analyses.** Fig A. To assess the influence of data revisions on our evaluation of forecast skill, we compared daily differences in cumulative reported cases during the week they were first reported to reported case counts for the same week in the complete data as of April 2, 2022. In total 721 weeks had at least one day with a revised case count (17% of all weeks, n = 4,241 weeks) and revisions occurred in 43 of 51 jurisdictions. These jurisdiction specific plots compare cases reported as of the date in the subtitle (in red) compared to cases reported as of April 2, 2022 (in black). Fig B. After identifying weeks with revised case counts, we then excluded them from the dataset and reran the GEE models and estimated the marginal mean Weighted Interval Score (WIS). Panel 1 shows the estimated marginal mean WIS and 95% confidence intervals for mean cases from team-specific GEE models for all 48 jurisdictions from this sensitivity analysis. The 95% confidence intervals for the COVIDhub-baseline model are shown in dashed red vertical lines. Panel 2 presents each team's estimated marginal mean WIS per phase, scaled to the COVIDhub-baseline model's estimated marginal mean WIS for all epidemic phases, using the dataset with excluded weeks. Teams with higher estimated marginal mean WIS values (i.e., greater than 1.0) are presented in shades of orange while teams with lower estimated marginal mean WIS (i.e., less than 1.0) are shown in shades of green. Team forecasts are denoted with an asterisk (*) if the 80% confidence interval of the expected WIS outcome (normalized and on the log scale) was estimated by a model to be lower than the average expected WIS of the COVIDhub-

baseline model across all phases. Fig C. Outliers were defined as non-revised reported case counts that were outside of the expected range by at least two of the three algorithms: a rolling median, a seasonal trend decomposition, and a seasonal trend decomposition without a seasonality term. Each method used a 21-day window. Approximately three percent of weeks (686 of 27,489 total weeks in the analysis period) had at least one day of reported cases identified as an outlier.
(PDF)

**S3 Appendix. Incident COVID-19 case forecasts were submitted for all US counties.** The plots shown here depicted average, scaled pairwise Weighted Interval Score (WIS; see *Methods* for description), 95% coverage, and submissions (Fig A), average 50%, 80% and 95% coverage for eligible submitted forecasts (Fig B), and average WIS and 95% coverage over time (Fig C). Each Fig shows spatial disaggregated results, with increasing population size and quintile numbers. For example, counties with the smallest population are grouped in Quintile 1 and the largest population sizes are grouped in Quintile 5. The following teams are included in these Figs: CEID-Walk, LNQ-ens1, Microsoft_DeepSTIA, COVIDhub-4_week_ensemble, COVID-hub-trained_ensemble, COVIDhub-baseline, CU-select, FAIR-NRAR, FRBSF_Wilson-Econometric, IowasStateLW-STEM, JHU_IDD-CovidSP, JHU_CSSE-DECOM, JHUAPL-Bucky, LANL-GrowthRate, LNQ-esn1, UVA-Ensemble. Fig A. Percent of weeks with complete submissions for all sets of team forecasts, scaled, pairwise relative Weighted Interval Score (rWIS), 95% coverage, and by geographical scale of submitted forecasts. Teams are sorted by increasing rWIS values. Fig B. Expected and observed coverage rates aggregated over time and horizon for county forecasts. The dashed line represents optimal expected coverage. Team forecasts that outperformed the COVIDhub-4_week_ensemble model at all coverage levels are labeled on the right hand side of the plots. Fig C. Mean Weighted Interval Score (WIS) over time, aggregated by geographic units and forecast horizon in A and 95% coverage over time, aggregated by geographic units and forecast horizon in B. The black, dashed vertical line in all panels shows the date that public communication of the case forecasts was paused. The black, dashed horizontal line in panels B show nominal 95% interval coverage.
(DOCX)

**S4 Appendix. Spatial correlation of forecast performance.** Fig A. Moran's I for each team's state-level relative Weighted Interval Score in the contiguous United States.
(DOCX)

**S5 Appendix.** Proportion of weeks in each classified epidemic phase (Fig A), and the estimated time-varying reproduction number and epidemic phase classifications (Fig B). Fig A. The proportion of weeks in each classified epidemic phase per state. Fig B. For each state, the top panel shows the median $R_t$ and median upper and lower 90% credible interval over time in red. The bottom panel shows reported case counts over time. Both plots have vertical bands representing the epidemic phase of each forecast week: *increasing*, *peak*, *decreasing*, *nadir*.
(PDF)

**S6 Appendix. Each location specific forecast submitted to the COVID19 Forecast Hub included at least 4 weeks of future predictions.** Here, we present disaggregated 1 and 4 week ahead predictions of model performance for each team model that submitted national and state/territory/DC forecasts and were included in the main analyses. Specific plots include the average 50%, 80% and 95% coverage for eligible submitted forecasts (Fig A), average absolute Weighted Interval Score (WIS) and 95% coverage over time (Fig B), and scaled, pairwise rWIS by location (Fig C) Fig A. Expected and observed coverage rates aggregated for 1 and 4 week ahead forecasts over time for national forecasts in 1, state/territory/DC forecasts in 2, the

largest county forecasts in 3. The dashed line represents optimal expected coverage. Teams that outperformed the COVIDhub-4_week_ensemble model at all coverage levels are labeled on the right-hand side of the plots. Fig B. Mean Weighted Interval Score (WIS) over time for 1 and 4 week ahead forecasts, aggregated by geographic units, and 95% coverage over time for 1 and 4 week ahead forecasts, aggregated by geographic units. The black, dashed vertical line in all panels shows the date that public communication of the case forecasts was paused. The black, dashed horizontal line in panels 3, 4, and 5 shows nominal 95% interval coverage. Teams that submitted national forecasts are presented in 1 and 4, state/territory/DC forecasts presented in 2 and 5, and teams that submitted large county forecasts are presented in 3 and 6. Fig C. Scaled, pairwise relative Weighted Interval Score (rWIS; see *Methods* for description) for all teams that submitted national and state/territory/DC forecasts by location for 1 and 4 week ahead horizon. National estimates are displayed first, followed by jurisdictions in alphabetical order. Teams are displayed by decreasing average rWIS across all forecast horizons and locations.
(DOCX)

**S7 Appendix. Phase- specific marginal mean Weighted Interval Score (WIS) over range of reported cases.** Fig A. Each team model's estimated marginal mean Weighted Interval Score (WIS) over a range of reported case counts per epidemic phase. Marginal mean WIS was estimated from GEE model results and reflects values across the 95% confidence interval of mean reported cases. Case counts differ per team model as each team forecasted a different set of locations over a different range of possible dates.
(DOCX)

**S8 Appendix. EPIFORGE 2020 guidelines outline 19 recommended reporting items for epidemic forecasting and prediction research (13).** These items are included in the checklist below, which also includes the page number where each item is described or presented within this evaluation.
(DOCX)

## Author Contributions

**Conceptualization:** Velma K. Lopez, Matthew Biggerstaff, Nicholas G. Reich, Michael A. Johansson.

**Data curation:** Estee Y. Cramer, Johannes Bracher, Alvaro J. Castro Rivadeneira, Aaron Gerding, Tilmann Gneiting, Yuxin Huang, Dasuni Jayawardena, Abdul H. Kanji, Khoa Le, Anja Mühlemann, Jarad Niemi, Evan L. Ray, Ariane Stark, Yijin Wang, Nutcha Wattanachit, Martha W. Zorn.

**Formal analysis:** Velma K. Lopez, Robert Pagano, John M. Drake, Eamon B. O'Dea, Madeline Adee, Turgay Ayer, Jagpreet Chhatwal, Ozden O. Dalgic, Mary A. Ladd, Benjamin P. Linas, Peter P. Mueller, Jade Xiao, Sen Pei, Jeffrey Shaman, Teresa K. Yamana, Samuel R. Tarasewicz, Daniel J. Wilson, Sid Baccam, Heidi Gurung, Steve Stage, Brad Suchoski, Lei Gao, Zhiling Gu, Myungjin Kim, Xinyi Li, Guannan Wang, Lily Wang, Yueying Wang, Shan Yu, Lauren Gardner, Sonia Jindal, Maximilian Marshall, Kristen Nixon, Juan Dent, Alison L. Hill, Joshua Kaminsky, Elizabeth C. Lee, Joseph C. Lemaitre, Justin Lessler, Claire P. Smith, Shaun Truelove, Matt Kinsey, Luke C. Mullany, Kaitlin Rainwater-Lovett, Lauren Shin, Katharine Tallaksen, Shelby Wilson, Dean Karlen, Lauren Castro, Geoffrey Fairchild, Isaac Michaud, Dave Osthus, Jiang Bian, Wei Cao, Zhifeng Gao, Juan Lavista Ferres, Chaozhuo Li,

Tie-Yan Liu, Xing Xie, Shun Zhang, Shun Zheng, Matteo Chinazzi, Jessica T. Davis, Kunpeng Mu, Ana Pastore y Piontti, Alessandro Vespignani, Xinyue Xiong, Robert Walraven, Quanquan Gu, Lingxiao Wang, Pan Xu, Weitong Zhang, Difan Zou, Graham Casey Gibson, Daniel Sheldon, Ajitesh Srivastava, Aniruddha Adiga, Benjamin Hurt, Gursharn Kaur, Bryan Lewis, Madhav Marathe, Akhil Sai Peddireddy, Przemyslaw Porebski, Srinivasan Venkatramanan, Lijing Wang, Pragati V. Prasad, Jo W. Walker, Alexander E. Webber.

**Methodology:** Velma K. Lopez, Estee Y. Cramer, Nicholas G. Reich, Michael A. Johansson.

**Project administration:** Estee Y. Cramer, Johannes Bracher, Alvaro J. Castro Rivadeneira, Aaron Gerding, Tilmann Gneiting, Yuxin Huang, Dasuni Jayawardena, Abdul H. Kanji, Khoa Le, Anja Mühlemann, Jarad Niemi, Evan L. Ray, Ariane Stark, Yijin Wang, Nutcha Wattanachit, Martha W. Zorn, Rachel B. Slayton, Matthew Biggerstaff, Nicholas G. Reich, Michael A. Johansson.

**Visualization:** Velma K. Lopez.

**Writing – original draft:** Velma K. Lopez, Estee Y. Cramer, Matthew Biggerstaff, Nicholas G. Reich, Michael A. Johansson.

**Writing – review & editing:** Velma K. Lopez, Estee Y. Cramer, Robert Pagano, John M. Drake, Eamon B. O'Dea, Madeline Adee, Turgay Ayer, Jagpreet Chhatwal, Ozden O. Dalgic, Mary A. Ladd, Benjamin P. Linas, Peter P. Mueller, Jade Xiao, Johannes Bracher, Alvaro J. Castro Rivadeneira, Aaron Gerding, Tilmann Gneiting, Yuxin Huang, Dasuni Jayawardena, Abdul H. Kanji, Khoa Le, Anja Mühlemann, Jarad Niemi, Evan L. Ray, Ariane Stark, Yijin Wang, Nutcha Wattanachit, Martha W. Zorn, Sen Pei, Jeffrey Shaman, Teresa K. Yamana, Samuel R. Tarasewicz, Daniel J. Wilson, Sid Baccam, Heidi Gurung, Steve Stage, Brad Suchoski, Lei Gao, Zhiling Gu, Myungjin Kim, Xinyi Li, Guannan Wang, Lily Wang, Yueying Wang, Shan Yu, Lauren Gardner, Sonia Jindal, Maximilian Marshall, Kristen Nixon, Juan Dent, Alison L. Hill, Joshua Kaminsky, Elizabeth C. Lee, Joseph C. Lemaitre, Justin Lessler, Claire P. Smith, Shaun Truelove, Matt Kinsey, Luke C. Mullany, Kaitlin Rainwater-Lovett, Lauren Shin, Katharine Tallaksen, Shelby Wilson, Dean Karlen, Lauren Castro, Geoffrey Fairchild, Isaac Michaud, Dave Osthus, Jiang Bian, Wei Cao, Zhifeng Gao, Juan Lavista Ferres, Chaozhuo Li, Tie-Yan Liu, Xing Xie, Shun Zhang, Shun Zheng, Matteo Chinazzi, Jessica T. Davis, Kunpeng Mu, Ana Pastore y Piontti, Alessandro Vespignani, Xinyue Xiong, Robert Walraven, Jinghui Chen, Quanquan Gu, Lingxiao Wang, Pan Xu, Weitong Zhang, Difan Zou, Graham Casey Gibson, Daniel Sheldon, Ajitesh Srivastava, Aniruddha Adiga, Benjamin Hurt, Gursharn Kaur, Bryan Lewis, Madhav Marathe, Akhil Sai Peddireddy, Przemyslaw Porebski, Srinivasan Venkatramanan, Lijing Wang, Pragati V. Prasad, Jo W. Walker, Alexander E. Webber, Rachel B. Slayton, Matthew Biggerstaff, Nicholas G. Reich, Michael A. Johansson.

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
