## [Decision Letter · Decision Letter 0]

6 Nov 2023

Dear Lopez,

Thank you very much for submitting your manuscript "Challenges of COVID-19 Case Forecasting in the US, 2020-2021" for consideration at PLOS Computational Biology.

I want to personally thank you for your patience during this lengthy review process. The authors are numerous, prolific, and highly collaborative, which made the search for highly qualified reviewers challenging! In general, all three reviewers provided positive reviews and constructive comments, and I look forward to reading a revised manuscript which addresses their concerns and thoughts. 

As with all papers reviewed by the journal, your manuscript was reviewed by members of the editorial board and by several independent reviewers. In light of the reviews (below this email), we would like to invite the resubmission of a significantly-revised version that takes into account the reviewers' comments.

We cannot make any decision about publication until we have seen the revised manuscript and your response to the reviewers' comments. Your revised manuscript is also likely to be sent to reviewers for further evaluation.

Sincerely,

Daniel B Larremore, Ph.D.

Academic Editor

PLOS Computational Biology

Thomas Leitner

Section Editor

PLOS Computational Biology

Reviewer's Responses to Questions

**Comments to the Authors:**

Reviewer #1: This paper analyzes the retrospective performance of COVID-19 case forecasts submitted to the forecast Hub. It describes the performance of these forecasts over time and across geographic units, and contrasts the performance of individual forecasts with that of ensemble models. It identifies that case forecasts were relatively unreliable (e.g., had low coverage rates for confidence intervals), particularly during periods of rapid increase, though ensemble models typically at least outperform simple baselines. It discusses directions that may be explored to improve future forecasting performance.

This is a much-needed and well executed paper. The fact that COVID case forecasts did not work well at key points during the pandemic is well known in the forecasting community, and a systematic attempt to document and understand this is an important part of moving the field forward. Overall, the analysis is quite thorough and most of my natural questions were answered in the supplemental material. The writing is clear throughout.

A couple of questions for the authors to consider:

(1) The results showed variation in forecast performance across locations. Is forecast performance spatially correlated? Either at the state level, or for counties within states? It would be quite interesting to see a choropleth of forecast skill and if there are properties of locations where forecasts performed better/worse. Beyond this, are residuals (as opposed to skill) spatially correlated? If so, this would argue in favor of models that incorporate explicit spatial structure, which is one of the recommendations made in the discussion section.

(2) Is there a measure of directional correctness for predictions? I.e., at the start of periods marked as increasing/decreasing, what fraction of the time did forecasters successfully guess the direction compared to the success rate of the baseline model? This would provide some concrete evidence about whether epidemic phase was "known" at the time to forecasters and they missed on magnitude, or whether there was also significant uncertainty about phase as well.

Minor typo: on line 360 there is a missing word in the phrase "clear association with forecast.."

Reviewer #2: Minor comments.

1. It is a shame that ignorance score (log likelihood) are not provided, since they are also proper scores and have the locality property. It could help in future systematic reviews of forecasting. Could the authors give us some idea about how a different choice of proper score might have changed any conclusions?

2. About 1/3 of the submitted forecasts were excluded, but it is not clear that this exclusion is justified. it would be worthwhile knowing how those other forecasts performed.

3. In the discussion, it is worth mentioning that some very influential forecasts or projections (call them what you will) at the beginning of the pandemic were worse than worthless. It is not necessary to mention the IHME projections by name, but their projection envelopes made in the spring of 2020 went down near zero for late summer 2020. People listened.

4. We don't get a full sense of why these forecasts were or were not performing well. One suspects that much of what was going on in the first year for sure was related to behavior-driven changes. People relented in their precautions, and a surge happened. People then became more careful. After the first year, vaccination and changes in immunity and viral evolution must have been important contributors.

5. It is hard to tell whether or not the data analysis fully adjusts for potential nonindependence of the outcomes. This needs to be very carefully explicated and made clear beyond doubt. (note line 563).

6. The paper says it contains all the code for figures and tables, but is there anything else in the results that involves unshared code?

Reviewer #3: ** General comments

This paper investigates case forecasts made in the US COVID-19 forecasting hub between July 2020 and December 2021. As similar previous and contemporary papers of this nature it is struggling to come to very firm conclusions on what determines performance but serves as an important record of these efforts, of the failure to make reliable forecasts (especially compared to death forecasts), and of performance by model, geography and time.

I was wondering if one of the main conclusions (forecasts performed worst in periods of rapid changes in reported cases, i.e. high absolute value of the reproduction number) is really borne out in the results. This statement is supported with two pieces of evidence: a decline in PI average and a greater marginal mean WIS compared to the baseline model. On the first (PI), there does indeed seem to be the relationship described but it is a bit hard to match on e.g. plots 1A and 6D when exactly coverage declines and which periods this correlates with. Perhaps this could be shown a bit clearer in a single plot, or more formally explored? For the greater marginal mean WIS compared to the baseline model, this is based on reproduction numbers that are used to define increasing/decreasing/turning phases. The turning phases should have reproduction number around 0, in which case I'd expect the baseline model to perform better than in the other phases. Is what we are seeing in the changes in marginal mean WIS compared to the baseline really the failure of the models to make good forecasts or is it the improved performance of the baseline? Perhaps it would be worth looking at how the marginal mean WIS values overall perform by phase, not just compared to the baseline?

The code repository needs some documentation. Also some of the code points to local files rendering the work non-reproducible. It would be great to fix this.

** Specific comments

l.114 and 118: I did not see how reference (11) shows that county-level case forecasts were used to inform vaccine trial selection (it very vaguely references both vaccine trial site selection and vaccine distribution but without spatial scale or evidence/description of what exactly happened), or how (13) shows that incorrect forecasts lead to erosion of trust. Can you be more specific in what aspects of these papers you are referring to here?

l.140 combines

l.172: "Three assumed that social distancing...changed during the prediction period." - can you be more specific on what distinguished these three models from the others?

l.238-240: performance of the baseline - similar to the reasoning above, are these jurisdictions where the trajectory was flatter? Are you really seeing varying quality of forecasts here or just variation in the shape of the epidemics?

l.248-249: seems to mix up rWIS and WIS, citing behaviour in the second as an example for behaviour int he first

l.252: "both metrics": which ones?

l.375: "forecasts of deaths have generally been more reliable": can you be more specific on how you quantify this?

l.399 "the e open"

l.419 "sable"

l.468-474 Inclusion criteria: can you give a rationale for these?

l.516 "We excluded...because they were not forecasted by most sets of team forecasts" is a strange sentence (do forecasts forecast? is forecasted a word?); also, can you give numbers?

l.530 I think you want https://github.com/epiforecasts/covid-rt-estimates

l.539: "Thus, we classified..." Does it matter in which order these operations were applied? E.g. what did you do if you had increasing-decreasing-increasing-decreasing?

l.570: please define all variables/symbols used in the equation

l.575: can you explain what exactly you mean by "formal inference" here and why it can't be done with rWIS values?

**Have the authors made all data and (if applicable) computational code underlying the findings in their manuscript fully available?**

Reviewer #1: Yes

Reviewer #2: Yes

Reviewer #3: **No: **Some of the code references files that are not available.

PLOS authors have the option to publish the peer review history of their article (what does this mean?). If published, this will include your full peer review and any attached files.

Reviewer #1: **Yes: **Bryan Wilder

Reviewer #2: No

Reviewer #3: **Yes: **Sebastian Funk

Figure Files:

Data Requirements:

Please note that, as a condition of publication, PLOS' data policy requires that you make available all data used to draw the conclusions outlined in your manuscript. Data must be deposited in an appropriate repository, included within the body of the manuscript, or uploaded as supporting information. This includes all numerical values that were used to generate graphs, histograms etc.. For an example in PLOS Biology see here: http://www.plosbiology.org/article/info:doi%2F10.1371%2Fjournal.pbio.1001908#s5.
---

## [Decision Letter · Decision Letter 1]

1 Apr 2024

Dear Lopez,

We are pleased to inform you that your manuscript 'Challenges of COVID-19 Case Forecasting in the US, 2020-2021' has been provisionally accepted for publication in PLOS Computational Biology.

Best regards,

Daniel B Larremore, Ph.D.

Academic Editor

PLOS Computational Biology

Thomas Leitner

Section Editor

PLOS Computational Biology

Thank you for your patience during the peer review process, as the breadth of the author list made finding reviewers especially challenging.

Reviewer's Responses to Questions

**Comments to the Authors:**

Reviewer #1: Thanks to the authors for their response. All of my concerns have been addressed and I'm happy for the paper to be accepted

Reviewer #3: Thank you. No further comments.

**Have the authors made all data and (if applicable) computational code underlying the findings in their manuscript fully available?**

Reviewer #1: None

Reviewer #3: Yes

PLOS authors have the option to publish the peer review history of their article (what does this mean?). If published, this will include your full peer review and any attached files.

Reviewer #1: **Yes: **Bryan Wilder

Reviewer #3: **Yes: **Sebastian Funk

---

## [Editor Report · Acceptance letter]

29 Apr 2024

PCOMPBIOL-D-23-00808R1 

Challenges of COVID-19 Case Forecasting in the US, 2020-2021

Dear Dr Lopez,

I am pleased to inform you that your manuscript has been formally accepted for publication in PLOS Computational Biology. Your manuscript is now with our production department and you will be notified of the publication date in due course.

With kind regards,

Anita Estes
